# Golden Buckwheat Extract–Loaded Injectable Hydrogel for Efficient Postsurgical Prevention of Local Tumor Recurrence Caused by Residual Tumor Cells

**DOI:** 10.3390/molecules28145447

**Published:** 2023-07-17

**Authors:** Li Xie, Rong Liu, Dan Wang, Qingqing Pan, Shujie Yang, Huilun Li, Xinmu Zhang, Meng Jin

**Affiliations:** 1School of Preclinical Medicine, Chengdu University, Chengdu 610106, China; xieli@cdu.edu.cn (L.X.); panqingqing@cdu.edu.cn (Q.P.);; 2Department of Pharmacy, Sichuan Nursing Vocational College, Chengdu 610100, China; wangdan1360@163.com; 3Department of Pharmacy, Chengdu University, Chengdu 610059, China; 4Clinical Medical College, Chengdu University, Chengdu 610106, China

**Keywords:** golden buckwheat extract, injectable hydrogel, localized chemotherapy

## Abstract

To prevent local tumor recurrence caused by possible residual cancer cells after surgery, avoid toxicity of systemic chemotherapy and protect the fragile immune system of postsurgical patients, an increasing amount of attention has been paid to local anti–cancer drug delivery systems. In this paper, golden buckwheat was first applied to prevent post–operative tumor recurrence, which is a Chinese herb and possesses anti–tumor activity. Golden buckwheat extract–loaded gellan gum injectable hydrogels were fabricated via Ca^2+^ crosslinking for localized chemotherapy. Blank and/or drug–loaded hydrogels were characterized via FT–IR, TG, SEM, density functional theory, drug release and rheology studies to explore the interaction among gellan gum, Ca^2+^ and golden buckwheat extract (GBE). Blank hydrogels were non–toxic to NIH3T3 cells. Of significance, GBE and GBE–loaded hydrogel inhibited the proliferation of tumor cells (up to 90% inhibition rate in HepG2 cells). In vitro hemolysis assay showed that blank hydrogel and GBE–loaded hydrogel had good blood compatibility. When GBE–loaded hydrogel was applied to the incompletely resected tumor of mice bearing B16 tumor xenografts, it showed inhibition of tumor growth in vivo and induced the apoptosis of tumor cells. Taken together, gellan gum injectable hydrogel containing GBE is a potential local anticancer drug delivery system for the prevention of postsurgical tumor recurrence.

## 1. Introduction

After surgical resection of the primary tumor, local recurrence and distant metastasis often occur due to residual tumor cells that go undetected [1,2]. Although chemotherapy is one of the most commonly used postoperative adjuvant therapies, chemotherapy–associated unsatisfactory clinical outcomes are prevalent. For instance, the oral bioavailability of chemotherapy drugs is generally low [3], the accumulation of drugs at the primary tumor site is relatively limited [4], it is difficult to reach and inhibit distant tumor cells [5], and it is easy to produce toxicity to healthy tissues or organs, leading to serious drug side effects [5,6]. In addition, after tumor surgery, the patient’s immune system is severely weakened. Therefore, systemic chemotherapy is usually initiated a few weeks post–surgery, allowing the recovery of the patient from surgery [7]. However, no drug administration is very likely to result in the continuous proliferation of residual invasive tumor cells, followed by tumor recurrence [7,8,9]; therefore, it is necessary for it to be administrated to patients immediately after tumor resection.

In recent years, an increasing amount of attention has been paid to the research, development, and application of local anti–cancer drug delivery system [10,11]. This delivery system can directly implant anti–cancer drugs into the surgical site to achieve accurate targeting. After local administration, the residual cancer cells are exposed to high concentrations of chemotherapy drugs for a long time and killed effectively, reducing the tumor recurrence rates [12,13,14,15]. In the meantime, the side effects are minimized due to the avoidance of the systemic circulation of chemotherapy drugs requiring high–dose oral or intravenous administration. For example, a carmustine–loaded biodegradable wafer (Gliadel^®^) is a local implant for post–surgical patients with glioblastoma [16]. In animal models, this implant in intracranial placement could locally release a high concentration of carmustine, which was increased by 4–1200 times, compared with that of intravenous administration [17]. Moreover, by comparison with placebo, Gliadel^®^ could prolong the overall survival of patients and had relatively low systemic toxicity, such as gastrointestinal diseases, fatigue, fever, and depression [11]. Consequently, the local delivery system of anti–cancer drugs is expected to be an alternative to systemic administration.

In China, traditional Chinese medicine (TCM) has been widely investigated in the field of cancer treatment. TCM prescriptions, proprietary Chinese medicine, the single Chinese herb, and Chinese herbal constituents have exhibited good anti–tumor activity. The case in point was Huaier granule, which could significantly prolong the recurrence–free survival of post–surgical patients with liver cancer and markedly reduced the rate of extrahepatic recurrence [18]. There are, of course, many other examples. For example, Xiaochaihu tang, Bupleuri radix, Radix Angelica or Cinnamomi cortex could improve the survival rate of tumor–bearing mice [19]. Many Chinese herbal constituents, such as curcumin, resveratrol, berberine, quercetin, and celastrol, among others, had anti–tumor effects on a variety of cancers [20]. In the above–mentioned research, the main administration routes of TCM are still oral administration and intravenous injection [21]. Few people have studied the local administration system of TCM as well as its effect on preventing post–operative tumor recurrence. Here, we made a bold attempt to introduce the single Chinese herb with anti–tumor effect into a local anti–cancer drug delivery system for post–operative administration of cancer. Not only could it meet the immediate administration of drugs after tumor resection, but it also had a relatively moderate pharmaceutical effect, which is more suitable for post–operative patients with a weak constitution.

Golden buckwheat (*Fagopyrum cymosum*) is a medicinal plant of the buckwheat family (Polygonaceae) collected in the Chinese Pharmacopoeia. Previous studies have shown that golden buckwheat has anti–cancer activities [22]. Flavonoids and phenols concentrated in the rhizome are the main anti–tumor active ingredients of golden buckwheat. The anti–tumor mechanism of these mainly included the inhibition of tumor cells’ proliferation and migration, the induction of tumor cells’ apoptosis and autophagy, as well as the anti–inflammatory and antioxidant effects. Most importantly, golden buckwheat belongs to the medicine food homology plant; therefore, it is a promising anti–tumor Chinese herbal medicine, whose effective components in the root were extracted and enriched to be used as a therapeutic drug in this paper. Additionally, in the process of fabricating a local anti–cancer drug delivery system for post–operative administration, how to solve the problem of filling irregular tumor resection cavity after surgery is extremely important. Hydrogels attracted a lot of attention due to their shape–adaptive function, and their resemblance to soft tissue [7,23,24]. Gellan gum is a linear polysaccharide obtained from a microorganism, *Pseudomonas elodea* [25], which is commonly applied as a food additive approved by USFDA and is also widely used in the field of biomedicine due to several advantages such as biocompatibility, non–cytotoxicity, biodegradability, gelation, low cost, and easy production [25,26,27]. Moreover, the properties of gellan gum can be easily regulated by calcium ions to prepare hydrogel materials. Thus, here, gellan gum injectable hydrogel was established by gellan gum and calcium ions via the electrostatic and hydrogen bond interactions, and was selected as the carrier of golden buckwheat extract (GBE) to construct the local drug delivery system for preventing postsurgical tumor recurrence caused by residual tumor cells (Figure 1). The effects of the amount of calcium ions on the characteristics of gellan gum injectable hydrogel, including morphology and thermal properties, were systematically investigated. The influence of drug loading content on release was also investigated through drug release studies and calculated via DFT. The injectability, flexibility and mechanical strength of blank hydrogel and GBE–loaded hydrogel were studied. The in vitro biocompatibility of blank hydrogel with different mixed ratios of gellan gum and calcium ions, and the in vitro anti–tumor effect of GBE and GBE–loaded hydrogels were evaluated on B16 and HepG2 cells. The hemocompatibility of blank hydrogel and GBE–loaded hydrogel was evaluated. Furthermore, the inhibitory effect of GBE–loaded hydrogel on residual tumor cells after surgery was investigated via mice bearing B16 tumor xenografts.

## 2. Results and Discussion

### 2.1. GBE Content Analysis

(—) –Epicatechin (EC) is a flavonoid compound from golden buckwheat [22,28]. Additionally, its content must be assessed according to the pharmacopoeia in China when golden buckwheat is used as raw medicinal material. In our laboratory, golden buckwheat extract was obtained from golden buckwheat root via ultrasonic method, whose yield was up to 20.73%. EC content in the obtained GBE was 0.202%. Furthermore, EC content in golden buckwheat root was determined to be 0.042% following calculations, which was much higher than the standard content of golden buckwheat in the 2020 edition of Chinese pharmacopoeia (no less than 0.020%). This proved that the chosen golden buckwheat was of good quality and the extraction method was reasonable and effective.

### 2.2. FT–IR Study of Blank Hydrogels

GG is composed of glucose, glucuronic acid and rhamnose in a molar ratio of 2:1:1, which is negatively charged owing to glucuronic acid containing a free carboxyl side group [29]. Its structure was characterized via FT–IR (Figure 1a). In the FT–IR spectrum, the characteristic absorption peak of GG at approximately 1404 cm^−1^ was ascribed to carboxylate symmetric stretching vibration of glucuronic acid. When Ca^2+^ was added into GG at COO^−^/Ca^2+^ molar ratios of 1:1, 1:4 and 1:8, the symmetric stretching vibration absorption peak of carboxylate of glucuronic acid appeared at 1416, 1425, and 1416 cm^−1^, respectively (Figure 1b–d). Obviously, the addition of Ca^2+^ resulted in red shifts larger than 10 cm^−1^ in the carboxylate symmetric stretching vibration peak. These red shifts revealed that ionic bonds were formed between carboxyl of GG and Ca^2+^, and blank hydrogels were successfully prepared [30].

### 2.3. TGA Study of Hydrogels

The thermal stability of hydrogels under a nitrogen atmosphere was investigated via thermogravimetric analyses. Figure 2 compared the TGA curves of GG, blank hydrogels with different COO^−^/Ca^2+^ molar ratios, GBE and 2.5% GBE–loaded hydrogel with the COO^−^/Ca^2+^ molar ratio of 1:1, which were obtained at a heating rate of 10 °C·min^−1^. Generally, in the 30–800 °C temperature range, the thermal decomposition of GG and blank hydrogels with the COO^−^/Ca^2+^ molar ratios from 1:1 to 1:8 could be divided into two main stages. The first stage corresponded to the temperature range of 30–150 °C, which was correlated with water evaporation, whereas in the second step, the main loss of mass observed could be due to polymer chain breakage. Moreover, the initial decomposition temperatures of blank hydrogels with the COO^−^/Ca^2+^ molar ratio from 1:1 to 1:8 were 245.62, 244.46, and 242.36 °C, respectively, which were slightly higher than that of GG (241.84 °C). It was indicated that there was an interaction between GG and Ca^2+^, which enhanced thermal stability for blank hydrogels. However, with increasing Ca^2+^ content, the decomposition temperature of blank hydrogels gradually decreased, suggesting that the interaction between GG and Ca^2+^ became weak, and excessive Ca^2+^ was not conducive to the formation of stable ionic bonds between GG and Ca^2+^.

For GBE, thermogravimetry showed two steps of weight loss within the temperature interval of 30–800 °C. After GBE was loaded into blank hydrogel with the COO^−^/Ca^2+^ molar ratio of 1:1, the obtained curves showed that the thermograms of 2.5% GBE–loaded hydrogel with the COO^−^/Ca^2+^ molar ratio of 1:1 were characterized by three distinct events. Thermal decomposition of GBE and 2.5% GBE–loaded hydrogel with the COO^−^/Ca^2+^ molar ratio of 1:1 was complicated, because GBE is a complex mixture with flavonoids, phenolics and so forth. Nevertheless, in terms of the initial decomposition temperatures, that of 2.5% GBE–loaded hydrogel with the COO^−^/Ca^2+^ molar ratio of 1:1, 154.63 °C, was higher than that of GBE, 140.65 °C. However, both of these were also lower than that of blank hydrogel with the COO^−^/Ca^2+^ molar ratio of 1:1, 245.62 °C. It was shown that the addition of GBE reduced the initial decomposition temperature of blank hydrogel with the COO^−^/Ca^2+^ molar ratio of 1:1, resulting in low thermal stability. 

### 2.4. Morphology Study of Hydrogels

Morphological analysis of freeze–dried hydrogels was performed via SEM, which was used to investigate the effect of Ca^2+^ and/or GBE on the morphology of hydrogels (Figure 3). When hydrogel was prepared via only GG in COO^−^/Ca^2+^ molar ratio of 1:0, it displayed irregular pore shapes with unclear boundaries. By adding Ca^2+^ into GG, a membrane–like structure was observed in the hydrogel, especially for blank hydrogel with a COO^−^/Ca^2+^ molar ratio of 1:1. Interestingly, there was an obvious difference in the membrane–like structure for different content of Ca^2+^ in GG. With increase in the Ca^2+^ content according to COO^−^/Ca^2+^ molar ratios from 1:0 to 1:8, irregular pore shapes disappeared and a membrane–like structure with few holes appeared in SEM images of hydrogels. This could be explained by addition of appropriate amounts of Ca^2+^ possibly prompting the crosslinking of the negatively ionized carboxylate groups on the GG chains [31], and stabilizing the double helix structure of GG [32], resulting in the formation of a membrane–like structure. However, excessive Ca^2+^ would be unfavorable for an increase in the overall orderly level of double helices of GG [31]. Thus, the membrane–like structure was distorted, and big holes were found in blank hydrogels with a COO^−^/Ca^2+^ molar ratio of 1:8. After GBE loading, the morphology of blank hydrogels was reshaped. The irregular–shaped sheet structure formed when GBE was added into GG in COO^−^/Ca^2+^ molar ratio of 1:0. Pores were detected in GBE–loaded hydrogel with a COO^−^/Ca^2+^ molar ratio of 1:1 and relatively smooth membrane–like structures appeared in drug–loaded hydrogels with COO^−^/Ca^2+^ molar ratios of 1:4 and 1:8. Compared with blank hydrogels, the observed morphological changes of GBE–loaded hydrogels were probably caused by the interaction between blank hydrogel and GBE [33,34]. 

### 2.5. Drug Release Studies of Hydrogels

For good thermal stability, blank hydrogel with a COO^−^/Ca^2+^ molar ratio of 1:1 was chosen as a drug delivery carrier to evaluate a controllable GBE–release property and the interaction between GBE and blank hydrogel. The in vitro release of GBE–loaded hydrogels with different drug loading content (1%, 2.5% and 5%) was determined via HPLC analysis. Release profiles of EC from GBE–loaded hydrogels with a COO^−^/Ca^2+^ molar ratio of 1:1 were shown in Figure 4A. In terms of 1% and 2.5% of drug loading content, GBE–loaded hydrogels with a COO^−^/Ca^2+^ molar ratio of 1:1 demonstrated a sustained release, with more than 90% within 12 h. Moreover, there was no absolute difference in the 12 h–cumulative release rate of EC (*p* > 0.05). Significantly, when drug loading content was up to 5% in the GBE–loaded hydrogels with a COO^−^/Ca^2+^ molar ratio of 1:1, the accumulated release rate of EC was less than 65% within 12 h, which was obviously different from the 1% GBE–loaded hydrogel and 2.5% GBE–loaded hydrogel (*p* < 0.001). This might be explained by the interaction between blank hydrogel and GBE preventing the entrapped EC from being released, which was further proven via DFT. According to the data presented in Figure 4B, when EC was introduced into blank hydrogel with a COO^−^/Ca^2+^ molar ratio of 1:1, by increasing the concentration of EC, the binding energy between Ca^2+^ and GG gradually weakened; however, that of GG and EC progressively strengthened. Hydrogen bonds could be formed between various hydroxyl groups on GG and components containing phenolic hydroxyl groups in GBE. 

### 2.6. Injectability, Flexibility and Mechanical Strength Studies of Hydrogels

The injectability of blank hydrogel/GBE–loaded hydrogel with a COO^−^/Ca^2+^ molar ratio of 1:1 from 1 mL syringe with three different types of curved needles (19 G, 20 G, and 22 G) was observed and recorded via images (Figure 5A). For blank hydrogel (COO^−^: Ca^2+^ = 1:1, mol/mol), it was present in the form of droplets during the process of injection, no matter which type of needle was used for injection. Moreover, when blank hydrogel was continuously pushed out from the syringe, the resistance force was relatively great. Compared with blank hydrogel, the injectability of 2.5% GBE–loaded hydrogels had a great change, which was injected into water in the form of a linear shape using three needles. In addition, in the injection process, the resistance force was small. The above–mentioned observations all showed that GBE could improve injectability of blank hydrogel, which was probably due to the interaction between blank hydrogel and GBE.

The flexibility and mechanical strength of hydrogels were characterized via a rheology study. As shown in Figure 5B, the storage modulus (G′) of blank hydrogel gradually decreased with increasing strain and intersected with the loss moduli (G″) at ~160% strain, suggesting the damage of the hydrogel. The critical strain of blank hydrogel with a COO^−^/Ca^2+^ molar ratio of 1:1 was greater than 100%, thus demonstrating good flexibility [35]. Loss factor (tan δ) is equal to the ratio of G″ to G′. In Figure 5C, tan δ of both blank hydrogel and 2.5% GBE–loaded hydrogel was less than 1 at angular frequencies in the range from 0.1 to 100 rad/s, which indicated that G′ was always larger than G″. The result showed that the two above–mentioned hydrogels maintained gel states. Furthermore, 2.5% GBE–loaded hydrogel showed a higher G′ than that in blank hydrogel in the whole range of frequencies studied (Figure 5D), suggesting that the introduction of GBE in hydrogel increased its mechanical strength. The possible mechanism of action was that more interaction existed between GBE and blank hydrogel. 

### 2.7. In Vitro Biocompatibility of Blank Hydrogels

To evaluate the cytotoxicity of blank hydrogels with different molar ratios of COO^−^ to Ca^2+^ (1:1, 1:4 and 1:8), cell viability in NIH3T3 cells was determined, which were incubated with the above materials for 24 h and 48 h. As shown in Figure 6A, there were no obvious differences in cell viability of blank hydrogels with COO^−^ to Ca^2+^ ratios of 1:1 and 1:4 (mol/mol) at any point in time, which all reached above 95%. For blank hydrogel with a COO^−^/Ca^2+^ molar ratio of 1:8, the cell viability was slightly reduced from 95% to 89% when the incubation time was prolonged from 24 h to 48 h. In addition, after 48 h of incubation, NIH3T3 cells were also examined microscopically to determine changes in general morphology. Our studies demonstrated that compared with the control group, most of the cells treated with all blank hydrogels exhibited good cell adhesion, stretched well, and had good morphology on the cell culture plate (Figure 6B). These results indicated that blank hydrogels consisting of GG and Ca^2+^ were minimally toxic and had good biocompatibility, which was consistent with the literature [36,37]. 

### 2.8. In Vitro Hemolysis Assay

Before the in vivo biological evaluation, the blood compatibility of blank hydrogel with a COO^−^/Ca^2+^ molar ratio of 1:1 and 2.5% GBE–loaded hydrogel with a COO^−^/Ca^2+^ molar ratio of 1:1 was assessed via a hemolytic assay utilizing mice blood. RBCs of mice were treated with the above–mentioned hydrogels for 3 h. The color of the supernatant and the hemolysis rates of water, PBS and hydrogel samples were illustrated in Figure 6C. Obviously, the color of the supernatant of blank hydrogel and 2.5% GBE–loaded hydrogel was similar to that of PBS. Moreover, the hemolysis rates of blank hydrogel and 2.5% GBE–loaded hydrogel were 0.1% and 3.9%, respectively, which met the international standards (ISO/TR 7405, the hemolysis rate of the blood–contacting material should be less than 5.0%) [38]. This clearly demonstrated that blank hydrogel and 2.5% GBE–loaded hydrogel did not display hemolytic activity for RBCs, which had good blood safety in the in vivo study.

### 2.9. In Vitro Anti–Tumor Evaluation

In vitro anti–tumor effects of GBE–loaded hydrogels were tested using a cytotoxicity assay. Simply, B16 and HepG2 cells were separately incubated with free GBE and GBE–loaded hydrogel containing a molar ratio of COO^−^ to Ca^2+^ of 1:1 for 48 h. The final concentrations of GBE in the culture medium were 0.15 µg/µL, 0.24 µg/µL, 0.3 µg/µL and 0.6 µg/µL, respectively. Cell viability was further assessed using an MTT assay, as shown in Figure 7. The results of the MTT assay showed that both free GBE and GBE–loaded hydrogels showed high cellular mortality at 48 h. On B16 cells (Figure 7A), the cell viability was less than 50% at the concentration of free GBE of 0.15 µg/µL, and was further reduced to lower than 20% at the concentration of free GBE of 0.6 µg/µL. It was shown that free GBE had very high anti–tumor activity on B16 cells. Compared with B16 cells, HepG2 cells were more sensitive to free GBE (Figure 7B). A total of 0.15 µg/µL free GBE inhibited the viability of HepG2 cells by approximately 10%. Furthermore, there was no significant difference in cell viability at the concentrations of free GBE from 0.15 to 0.6 µg/µL. More importantly, the cell mortality of GBE–loaded hydrogels was similar to that of free GBE on both B16 and HepG2 cells. That suggested that the anti–tumor active components of GBE could be almost completely released from hydrogels. In the same way, the above–mentioned results could be observed from the optical images of the tested cells (Figure 7C,D). In the low dosage group (0.15 µg/µL) for either free GBE or GBE–loaded hydrogels, approximately half of the B16 cells maintained cellular morphology in comparison with the control group. In addition, by increasing the dosage from 0.15 to 0.6 µg/µL, fewer and fewer cells were visible in the images, while for HepG2 cells, almost no normal cells could be found in all dosages of GBE–loaded hydrogels and free GBE. 

### 2.10. In Vivo Anti–Tumor Evaluation

The in vivo inhibitory effect of GBE–loaded hydrogel on local tumor recurrence caused by residual tumor cells after surgery was evaluated in C57BL/6 mice bearing B16 murine melanoma xenograft (Figure 8). In order to carry this out, a microtumor with grain size was firstly left in the tumor resection cavity after surgery to provide a chance of local tumor recurrence. Thereafter, 2.5% GBE–loaded hydrogel with a COO^−^/Ca^2+^ molar ratio of 1:1 was chosen to be injected into the tumor resection cavity of mice owing to its excellent anti–tumor effect in vitro. According to the experiment design (Figure 8A), after one injection, the mice were fed for 14 days, their body weights were monitored every other day and post–surgical tumor recurrence was detected on the last day. During the experiment, there was a modest increase in body weight of mice, but no significant difference among the control group, the blank hydrogel group and the GBE–loaded hydrogel group (Figure 8B), indicating the low toxicity of GBE and hydrogel in vivo. In addition, in the tumor resection cavity, residual microtumors underwent regrowth in all three groups. Nevertheless, the recurrent tumor in GBE–loaded hydrogel group was slightly smaller than that in the control and blank hydrogel groups (Figure 8C). The killing effect of GBE–loaded hydrogel on recurrent tumor cells was also determined via histological and immunohistochemical studies (Figure 8D). Based on H&E staining, the recurrent tumor tissues in the control and blank hydrogel groups displayed a normal tumor section morphology, such as abundant blood vessels, compact tumor cells and rich chromatin in the image. In contrast, topical administration of GBE–loaded hydrogel made recurrent tumor tissue become loose and led to the emergence of large areas of necrosis, and most tumor cells were dead from apoptosis with broken pieces and missing nuclei. Meanwhile, TUNEL assay was further performed to detect apoptosis in situ. It was apparent that many TUNEL–positive cells were found only in recurrent tumors of GBE–loaded hydrogel group, which indicated that GBE–loaded hydrogel efficiently induced apoptosis in tumor cells. Ultimately, Ki67 antigen staining was used to reveal the activity of proliferation of tumor cells. Tumor sections in both the control group and the blank hydrogel group had remarkable brown staining and more negative blue staining than that in GBE–loaded hydrogel group. The above–mentioned results demonstrated that GBE–loaded hydrogel had the ability to kill residual tumor cells after surgery or recurrent tumor cells in vivo.

## 3. Materials and Methods

### 3.1. Materials and Reagents 

Food–grade low acyl gellan gum (GG) was purchased from Wuhan Wanrong Technology Development Co., Ltd. (Wuhan, China, CAS NO: 71010-52-1). Its average molecular weight (M_w_) and purity were 500 kDa and 99.6%, respectively. Golden buckwheat roots were purchased from Anhui Daoyuantang Decoction Pieces Co., Ltd. (Bozhou, China). Anhydrous calcium chloride and antibiotics (penicillin and streptomycin) were obtained from Sigma-Aldrich Co. (Steinheim, Germany). The methyl thiazolyl tetrazolium (MTT) cell viability assay kit was obtained from the Beyotime Institute of Biotechnology (Shanghai, China). Dulbecco’s modified Eagle’s medium with high glucose (DMEM), RPMI–1640 and fetal bovine serum (FBS) were obtained from Life Technologies Corporation (Gibco, ThermoFisher Scientific, Paisley, UK). The buffers were prepared in MilliQ ultrapure water and filtered (0.22 mm) prior to use, and all other materials and reagents were of analytical grade and purchased from Kelong Chemical Co. (Chengdu, China). A murine melanoma cell line (B16), a human hepatoma cell line (HepG2) and a mouse embryonic fibroblast cell line (NIH3T3) were obtained from the Shanghai Institutes for Biological Sciences (Shanghai, China). 

### 3.2. Methods

Fourier transform infrared (FT–IR) spectra were recorded on FT–IR Spectrometer Spectrum 3 in the wave number range of 4000–400 cm^−1^. Thermal analysis was performed on a simultaneous thermogravimetry/differential scanning calorimetry instrument (TGA/DSC, TGA/DSC 3+, METTLER TOLEDO, Zurich, Switzerland). The thermal decomposition of freeze–dried samples was carried out under a nitrogen atmosphere from 30 to 800 °C at a heating rate of 10 °C·min^−1^. The morphology of samples was characterized using a scanning electron microscope (SEM, JEM–IT300, JEOL, Tokyo, Japan). The rheological behavior of samples was characterized using a rheometer (MCR302e, Anton Paar, Graz, Austria). All tests were carried out at 30 °C with a stainless–steel parallel plate (50 mm diameter; gap: 1 mm). The strain amplitude sweeps were conducted with a strain amplitude in the range from 0.1% to 1000% at the angular frequency of 10 rad/s. Oscillatory frequency sweeps were performed at an angular frequency ranging from 0.1 to 100 rad/s at 0.3% strain. 

### 3.3. Preparation of Golden Buckwheat Extract (GBE)

The dried roots of golden buckwheat were crushed into powder, passed through a 20-mesh sieve, soaked overnight in 50% ethanol, and extracted three times with 10-fold 50% ethanol through ultrasonic assistance (Ultrasonic Cell Crusher JY92–IIDN from Ningbo Scientz Biotechnology Co., Ltd., Ningbo, China) at 80% power for 30 min. Then, the supernatant was filtered twice using a sand core funnel with G3 pore size, and ethanol was removed with a vacuum rotary evaporator. Finally, GBE was obtained after lyophilization. The yield of it was calculated as the following Formula (1):Yield of GBE (%) = (GBE weights/golden buckwheat roots powder weights) × 100(1)

### 3.4. GBE Content Analysis

According to the Chinese pharmacopoeia 2020 edition, epicatechin (EC) was selected to characterize the content and the release profile of GBE using high–performance liquid chromatography (HPLC, Shimadzu LC-20A, Shimadzu, Kyoto, Japan). Briefly, a certain amount of GBE (10 mg) was added to 50% ethanol (10 mL) and treated via ultrasonic for 3 min to completely dissolve EC and centrifugation at 12,000 rpm for 5 min. The supernatant was withdrawn for HPLC assay, which was repeated twice. HPLC chromatographic conditions were as follows: Inertsil ODS-3 C18 chromatographic column (5 µm, 4.6 mm inner diameter × 250 mm; GL Sciences Inc, Tokyo, Japan); acetonitrile—0.004% phosphoric acid aqueous solution (10/90, *v*/*v*) as the mobile phase; the flow rate of 1.0 mL/min; column temperature 35 °C; detection wavelength of 280 nm; sample quantity 10 μL.

### 3.5. Preparation of Hydrogels

The preparation of 2 g hydrogel was taken for example. Briefly, 1 g GG aqueous solution (1%, *w*/*w*) was preheated to 90 °C. Under magnetic stirring at 600 rpm/min, 1 mL of different concentrations of calcium chloride aqueous solution (15.48, 61.92, and 123.84 mmol/L) was separately added dropwise to the GG solution to obtain the desired molar ratio of carboxylate group in each tetrasaccharide repeating unit of GG polymer to calcium ion (e.g., COO^−^: Ca^2+^ = 1:1, 1:4 or 1:8, mol/mol). After stirring for 30 min, the mixing solution formed a blank hydrogel when the temperature dropped to room temperature. For GBE–loaded hydrogels, a 0.2 mL GBE ethanol solution with various concentrations was slowly added to the mixing solution containing GG and calcium chloride. Subsequently, the above–mentioned solution was stirred for 10 min to form GBE–loaded hydrogel with 0.625%, 1%, 1.25%, 2.5% or 5% (*w*/*w*) of drug content. In the process of preparing hydrogels, the weight loss of hydrogels due to heating was adjusted with MilliQ ultrapure water.

### 3.6. Drug Release Studies of Hydrogels

Briefly, 2 g of GBE–loaded hydrogel was prepared in 20 mL glass vial. Drug release was performed in a shaking water bath of 37 °C. The shaking speed was maintained at 150 rpm. Then, 3 mL of phosphate–buffered solution (PBS, pH7.4) as the release medium was added to the glass vial. At set intervals, the release medium was withdrawn and replaced with an equal volume of fresh medium at 37 °C. The experiments were conducted in triplicate. Before the HPLC assay, the release medium was centrifugated at 12,000 rpm for 5 min. Then, the concentration of EC in the release medium was assayed and the cumulative release was calculated according to the following Equation (2).
(2)$$E\;(\%)=V_{t}\sum_{1}^{n}C_{t}/m_{0}\times 100$$
where, *E* is the cumulative release amount (%); *V_t_* is the sampling volume (3 mL); *C_t_* is the drug concentration (µg/mL); *t* is the sampling time (h); and *m*_0_ is the mass of EC in gel (µg). The results are expressed as the mean ± SD.

### 3.7. DFT Calculation

All density functional theory (DFT) calculations were carried out using Gaussian 16 program package [39]. Geometry optimizations were carried out to locate all of the stationary points by applying the M06–2X functional method [40,41] with the 6–31G(d) basis set [42,43] for all atoms. A polarized continuum model based on solute electron density (PCM–SMD) [44,45] was applied to simulate the solvent effect of aqueous solution. Moreover, the stability of the DFT wave–function of the auxiliary Kohn–Sham determinant was examined [46]. Harmonic vibrational frequency calculations were conducted to characterize all stationary point. Unless otherwise specified, the relative Gibbs free energies (ΔG, kJ mol^−1^) were obtained at the M06–2X/6–31G(d) − (PCM–SMD, water) level of theory. 

### 3.8. Injectability of Hydrogels

Blank hydrogel and GBE–loaded hydrogel with 2.5% (*w*/*w*) of drug content were chosen for qualitative evaluation of injectability. Hydrogels with a COO^−^/Ca^2+^ molar ratio of 1:1 were prepared according to “3.5 Preparation of Hydrogels”. When the hydrogel was present in a sol state, it was loaded in a 1 mL syringe. After gelation, the hydrogel in the 1 mL syringe was injected by using 19 G, 20 G and 22 G curved needles with blunt ends, respectively. To facilitate observation, blank hydrogel was directly injected into air because it was colorless and transparent, and GBE–loaded hydrogel was injected into water due to its orange–red color. 

### 3.9. In Vitro Biocompatibility of Blank Hydrogels

The in vitro biocompatibility of blank hydrogel was determined via the MTT assay on NIH3T3 cells. The cells were maintained in DMEM complete culture medium supplemented with 10% FBS and 1% penicillin–streptomycin and kept in a 37 °C humidified incubator with 5% CO_2_. Subsequently, cells were seeded in 24-well plates at a density of 2 × 10^4^ cells per well and cultured for 24 h. The culture media were refreshed with the fresh complete culture medium (1 mL). Meanwhile, 24 mg blank hydrogel with different molar ratios of carboxyl groups to calcium ions (1:1, 1:4, and 1:8) was also added into each well. Either 24 h or 48 h later, the blank hydrogel was removed from each well and the cell viability was evaluated using the MTT Cell Viability Assay Kit. The ultraviolet absorption of the dissolved formazan crystals at 490 nm was measured on a microplate reader (Synergy H1, BioTek, Winooski, VT, USA). Data are shown as the percent cell viability relative to control cells treated with PBS (pH 7.4) (*n* = 6).

### 3.10. In Vitro Hemolysis Evaluation

In vitro hemolysis test was performed according to previously described methods [47,48]. The hemocompatibility of blank hydrogel with a COO^−^/Ca^2+^ molar ratio of 1:1 and 2.5% GBE–loaded hydrogel with a COO^−^/Ca^2+^ molar ratio of 1:1 was evaluated. A total of 1.5 mL of whole blood was obtained from C57BL/6 mice. Red blood cells (RBCs) were separated via centrifugation at 3000 rpm for 5 min, washed with PBS for three times and diluted with PBS to 15 mL. Then, 3 mL of PBS and 0.6 mL of RBCs suspension were added into 20 mL glass vial containing 2 g of blank hydrogel or 2.5% GBE–loaded hydrogel as experiment control. In total, 3 mL of water and 3 mL of PBS were mixed with 0.6 mL of RBCs suspension as the positive and negative control, respectively. All groups were estimated from three parallel experiments. The samples were then incubated at 37 °C for 3 h. After incubation, the suspension in the glass vial was centrifuged at 3000 rpm for 5 min. The supernatant was added into a 96-well plate to measure the absorption value at 540 nm using an ultraviolet–visible light spectrophotometer (UV-7504, Shanghai Xinmao Instrument Co., Ltd., Shanghai, China). The hemolysis rate was calculated as in the following Formula (3):Hemolysis (%) = (A value of the experimental group − A value of the negative control)/(A value of the positive control − A value of the negative control) × 100%(3)

### 3.11. In Vitro Anti–Tumor Evaluation

The in vitro anti–tumor activity of GBE hydrogel was assessed using an MTT assay on B16 and HepG2 cells, which were separately cultured in RPMI–1640 and DMEM complete culture media in a 37 °C humidified incubator with 5% CO_2_. Cells were seeded in 24-well plates at a density of 2 × 10^4^ cells per well. A total of 24 h later, the culture media were removed and replaced with a fresh complete culture medium. Cells were, respectively treated with 100 µL GBE solutions and 24 mg GBE–loaded hydrogels (COO^−^: Ca^2+^ = 1:1, mol/mol) with different drug loading (0.625%, 1%, 1.25% and 2.5%, *w*/*w*) to make the final concentration of GBE in the culture medium reach 0.15 µg/µL, 0.24 µg/µL, 0.3 µg/µL and 0.6 µg/µL. After incubation for 48 h, the culture medium and GBE–loaded hydrogels were removed. The cells were rinsed with PBS (pH 7.4) and were observed using an inverted optical microscope (Zeiss AxioVert. A1, Carl Zeiss, Oberkochen, Germany). The cell viability detection and data processing were the same as in “3.9. In Vitro Biocompatibility of Blank Hydrogel”.

### 3.12. In Vivo Anti–Tumor Evaluation

Mice models with incompletely resected tumor were applied to estimate the inhibition effect of GBE–loaded hydrogel on residual tumor cells after surgery. All animal experiments followed the national guidelines for the Care and Use of Laboratory Animals. The male C57BL/6 mice (6-week old, Jian yang Experimental Animal Centre, Chengdu, China) were used to create subcutaneous tumors by injecting them with B16 cells (2 × 10^6^ cells per mice). The tumor excision was not performed until the tumors reached a diameter of ~10 mm. Firstly, the mice were anesthetized with 4% chloral hydrate (10 mL/kg, i.p.). Part of the tumor tissue was resected and tumor tissue with the size of a rice grain was intentionally left for the in situ simulation of a residual microtumor. Secondly, the hydrogel in a sol state was loaded in a 1 mL sterile syringe with a 22 G needle for injectable use. Thirdly, the fifteen post–surgical mice were randomly divided into three groups as follows: (1) the control group treated with nothing; (2) the blank hydrogel group treated with blank hydrogel (COO^−^: Ca^2+^ = 1:1, mol/mol) at a dose of 20 g/kg; and (3) the GBE–loaded hydrogel group treated with GBE–loaded hydrogel at a dose of 20 g/kg, which had a COO^−^/Ca^2+^ molar ratio of 1:1 and a drug loading of 2.5% (*w*/*w*). All the gels were locally injected only once at the residual microtumor site and the skin of the mice was then sutured after administration. The body weights of the mice were recorded every other day. Additionally, the mice were euthanized with isoflurane on the 14th day, and the tumors were excised. The recurrent tumor tissue samples were fixed with 4% paraformaldehyde solution and embedded in paraffin to cut sections with a thickness of 5 μm for histological analysis. The tissue sections were stained with hematoxylin and eosin (H&E), terminal deoxynucleotidyl transferase–mediated dUTP nick–end labeling (TUNEL) and Ki67, respectively, and imaged with a digital microscope (Moticam 2000, Motic, Xiamen, China).

### 3.13. Statistical Analysis

All biological data are expressed as means ± standard deviation (SD). Statistical analysis was performed using a single–factor ANOVA test. Datasets were compared using two–tailed, unpaired–tests. Values of *p* < 0.05 were considered statistically significant.

## 4. Conclusions

GBE–loaded gellan gum injectable hydrogel was successfully constructed, which is a local delivery system containing Chinese herbal medicine with anti–tumor activity. In the above–mentioned hydrogel, ionic bonds and hydrogen bonds were formed among GG, calcium ions and GBE, which resulted in significant differences in the accumulated release rate of EC within 12 h between 5% GBE–loaded hydrogel and 1% or 2.5% GBE–loaded hydrogel. The most significant aspect was that blank hydrogel and GBE–loaded hydrogel all possessed good blood compatibility. Both in vitro and in vivo anticancer activity of GBE–loaded gellan gum injectable hydrogel exhibited therapeutic efficacy of inhibiting the proliferation of tumor cells. These results indicate that the local anti–cancer delivery system of Chinese herbal medicine has therapeutic potential in inhibiting local tumor recurrence caused by residual tumor cells after surgery, which is worthy of further research.

## Data Availability

Not applicable.

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
