# Peer review of "Golden Buckwheat Extract–Loaded Injectable Hydrogel for Efficient Postsurgical Prevention of Local Tumor Recurrence Caused by Residual Tumor Cells"

_molecules, 2023, doi:10.3390/molecules28145447_

Round 1
Reviewer 1 Report
This research stands out from other works due to its novel approach to the problem of postoperative tumor recurrence. The use of traditional Chinese medicine in the form of golden buckwheat and the development of gellan gum hydrogels for local delivery of the anti-cancer drug represents a valuable combination of scientific knowledge with medical practice.
The results presented in the study indicate significant potential for gellan gum hydrogels containing GBE in preventing postoperative tumor recurrence, which may contribute to the development of new cancer therapies. Further research into the application of this technology may lead to enhanced oncological treatment efficacy and improved quality of life for patients.
For these reasons, I recommend this work for publication in this journal. The research presented holds great importance for medicine and may contribute to the development of new, more effective anti-cancer therapies.
Author Response
Thank you for your advice. We will conduct further research on local delivery systems loaded with traditional Chinese medicine with antitumor effect.
Reviewer 2 Report
Focal drug delivery is a promising approach to tri cancer as it decreases the unwanted systemic effect and enhances the drug effect via increasing its localized concentration. The manuscript is well-written an scientifically sound. My main critique is the misinterpretation of the results by the authors. Although the in vitro toxicity of the Golden buckwheat extract seems significant, the in vitro results have shown a very mild effect. Surprisingly, the authors concluded and stated in the abstract that “GBE has great potential for the prevention of postsurgical tumor recurrence” which is not supported by the data. The results clearly highlighted GBE's different anticancer activity behavior in vitro and in vivo scenarios. The in vivo activity is suppressed, and GBE application in vivo is questionable. This is an important observation worth publishing as it is common for many drugs.
There are other minor issues that need to be fixed prior the publishing.
1. More references need to be added related to different approaches I focal anticancer drug delivery, including common anticancer drugs such as: https://doi.org/10.1021/acsbiomaterials.6b00495
2. References 38 and 39 have the same title? All references must be double-checked.
Round 2
Reviewer 2 Report
The phrase in the abstract in line 23 "it efficiently inhibited the proliferation of tumor," should be changed to "it showed inhibition of tumor growth" . The efficiency and efficacy of in vivo antitumor activity of GBE is very moderate. I guess it could be used in a combinatory approach, but more studies are necessary.
Author Response
Thank you for your advice. The phrase in the abstract in line 23 "it efficiently inhibited the proliferation of tumor," has been changed to "it showed inhibition of tumor growth". The modified phrase in the revised manuscript has been marked in red.